# Microscopic Swarms: From Active Matter Physics to Biomedical and Environmental Applications

**DOI:** 10.3390/mi13020295

**Published:** 2022-02-13

**Authors:** Yulei Fu, Hengao Yu, Xinli Zhang, Paolo Malgaretti, Vimal Kishore, Wendong Wang

**Affiliations:** 1University of Michigan—Shanghai Jiao Tong University Joint Institute, Shanghai Jiao Tong University, Shanghai 200240, China; yulei.fu@sjtu.edu.cn (Y.F.); yuhengao2000@sjtu.edu.cn (H.Y.); zhxl304284740@sjtu.edu.cn (X.Z.); 2Helmholtz Institute Erlangen-Nürnberg for Renewable Energy (IEK-11), Forschungszentrum Jülich, 52425 Jülich, Germany; p.malgaretti@fz-juelich.de; 3Department of Physics, Banaras Hindu University, Varanasi 221005, India; vimalk@bhu.ac.in

**Keywords:** microscopic swarms, active matter, collective behavior, microrobots

## Abstract

Microscopic swarms consisting of, e.g., active colloidal particles or microorganisms, display emergent behaviors not seen in equilibrium systems. They represent an emerging field of research that generates both fundamental scientific interest and practical technological value. This review seeks to unite the perspective of fundamental active matter physics and the perspective of practical applications of microscopic swarms. We first summarize experimental and theoretical results related to a few key aspects unique to active matter systems: the existence of long-range order, the prediction and observation of giant number fluctuations and motility-induced phase separation, and the exploration of the relations between information and order in the self-organizing patterns. Then we discuss microscopic swarms, particularly microrobotic swarms, from the perspective of applications. We introduce common methods to control and manipulate microrobotic swarms and summarize their potential applications in fields such as targeted delivery, in vivo imaging, biofilm removal, and wastewater treatment. We aim at bridging the gap between the community of active matter physics and the community of micromachines or microrobotics, and in doing so, we seek to inspire fruitful collaborations between the two communities.

## 1. Introduction

Collective behavior is ubiquitous in natural and artificial systems across all scales, ranging from the macroscopic, such as bird flocks [1], fish schools [2,3], mammal herds [4,5], ant colonies [6], and marching locusts [7] to the microscopic, such as bacteria colonies [8], molecular motors [9,10], autophoretic colloids [11], and microrobotic swarms [12]. Although the length scales and cognitive abilities of constituent individuals are different for these systems, they all belong to the category of active matter system. Individuals in these systems consume the free energy produced either within themselves or from their surroundings to perform mechanical work. Thus, these systems share some common self-organizing phenomena. Nevertheless, unlike the static equilibrium self-assembly, the active nonequilibrium self-organization is still poorly understood and presents many great challenges and opportunities [13]. Existing theories on active matter have described some characteristic cases and predicted a few striking phenomena [14,15,16].

Robotics is a crucial part of industry 4.0, which signals the intelligent era of industry and human civilization. Micro/nanorobotics is an emerging field that borrows the concepts from both nano- and micro-technology and robotics. Because micro/nanorobots are too small to have circuits integrated and printed on them, it is difficult to control them the same way as we control macroscopic robots. So far, a single micro/nanorobot was manipulated mainly by external fields, such as magnetic or acoustic fields, and has been relatively widely used in reality, such as in the medical, environmental protection, and other engineering fields [17,18,19,20,21,22]. However, because of the small size of a single micro/nanorobot, its functions are limited, so we want to utilize the micro/nanorobotic swarms to enhance their ability and to develop a broad range of applications such as drug delivery or sewage treatment. Hence, the research on the emergent collective behaviors of micro/nanorobots is an emerging field of research [23,24,25,26,27,28].

We want to show the significance of emergent collective behaviors from both the perspective of fundamental physics of active matter systems and from the perspective of the application of micro/nanorobotic swarms. Here, we use the term microscopic swarms to refer to particle-based active matter systems as well as micro/nanorobotic swarms. As shown in Figure 1, we first introduce the fundamental studies on the physics of some unique phenomena of active matter systems, such as long-range order, giant number fluctuation, motility-induced phase separation, and our recent work on the relationship between information and order in a self-organizing driven system. Next, we discuss the applications of microscopic swarms. We start with swarm manipulation methods, such as magnetic or electric fields, light, acoustic waves, or chemicals, and briefly discuss the pros and cons of different methods. Then we introduce some specific application areas such as target therapy, in vivo imaging, biofilm removal, and environmental treatment. Finally, we summarize the current limitations and envision future directions of the microscopic swarms.

## 2. The Perspective of Fundamental Physics

Active matter systems show very different properties and phenomena compared with equilibrium systems due to the violation of time-reversible symmetry and the principle of detailed balance [29]. Many theoretical models in this field were proposed first and confirmed by experiments. This section combines the research results from both theoretical and experimental communities. We will briefly discuss the theoretical origin of each property and phenomenon and then describe the experimental evidence that fully or partly corroborated the theoretical predictions.

### 2.1. Long-Range Order

Collective motion in two dimensions (2D) with long-range orientational order occurs in active systems. In contrast, equilibrium systems in 2D do not exhibit long-range order, according to the Mermin–Wagner theorem [30]. From a theoretical perspective, the seminal work of Vicsek et al. first brought the flocks into the broad category of active matter systems [31]. They built a simple discrete-time, discrete-element model partly based on the Reynold’s assumption, which states that the velocity of each bird is only affected by the local dynamic environment and its interaction with neighbors [32]. Their simulation result predicts that phase transition from disorder to long-range orientational order (so-called polar order) occurs in polar particles when the noise (temperature) is reduced below the critical value. Afterward, in the 1990s, Toner and Tu et al. proposed a nonequilibrium hydrodynamic model for the collective motion of “ferromagnetic” flocks based on considerations of the symmetries and conservation laws, similar to Navier–Stokes equations [33,34,35]. This coarse-grained continuum field model further confirmed the existence of the long-range order.

Recent experiments show that the long-range orientational order exists in the microscopic swarms. First, we describe the examples in living systems. Nishiguchi et al. studied long filamentous bacteria in a thin fluid layer, and for a large enough density they observed long-range nematic order due to the collision (Figure 2a) [36]. The nematoid arrangement was also found in the cultures of migrating and interacting human cells such as melanocytes, adipocytes, osteoblasts, etc., and the arrangement was formed by apolar interaction (Figure 2b) [37,38].

Besides biological systems, long-range orientational order was observed in artificial systems too. Deseigne et al. studied a monolayer of vibrated millimeter-scale polar disks and observed the alignment motion during collision (Figure 2c) [39,42]. Bricard et al. reported that dilute colloidal particles propelled by an electrohydrodynamic mechanism called a Quincke rotation could self-organize to utilize the reorientational movement to achieve emergent directed motion (Figure 2d) [40]. The long-range orientational order also occurs in 2D microfluidic droplets, where the velocity correlation of micro-droplets exhibit power-law decay, which suggests that the velocity shows quasi-long range order (Figure 2e) [41].

Although the long-range directional order phenomenon is ubiquitous in self-propelled agents, the cause of this phenomenon is complex and diverse. For non-living systems such as colloids or droplets, it may be due to the symmetry of the two-body interaction (attractive/repulsive force or inelastic collision) and the kinematics of particles [43,44,45]. Meanwhile for living systems, such as bird flocks or bacteria colonies, quorum sensing may also exist, caused by the external gradient (e.g., chemical, magnetic, etc.) that helps individuals to align with others, thus generating the collective motion [39].

### 2.2. Giant Number Fluctuation

One characteristic phenomenon of the active matter systems is giant number fluctuation (GNF) [46]. Briefly, the number of active particles (N) in a selected sub-volume (V) changes with time, and the root mean square number fluctuation of particles $\sqrt{\langle \delta N^{2}\rangle }$ (notated as ΔN), where δN≡N−〈N〉, should be proportional to $\sqrt{N}$ in almost all equilibrium systems and in most nonequilibrium systems due to the law of large numbers [47]. However, the number fluctuation in some active matter systems, particularly the ones with orientational order, is significantly larger than in other systems, which obey the law of large numbers. Suppose we use $\Delta N\propto {\langle N\rangle }^{\alpha }$ to express the relationship between the standard deviation and the average number of particles; the exponent in GNF could be as large as 1, in contrast with ½ in equilibrium systems. Due to its ease of measurement, GNF has been reported in many experiments, including millimeter-scale particles, colloidal particles, bacteria, and cells (Table 1).

GNF was first predicted in dry (i.e., non-momentum conserving) apolar active fluids (nematic order) [58], and was then extended to the dry polar active fluids (ferromagnetic flocks) [59] based on hydrodynamic equations derived from symmetry and conservation laws [15]. According to the physical model, the GNF exponent for nematic (apolar) alignment interaction [47] is $\alpha =\frac{7}{10}+\frac{1}{5D}$ and for ferromagnetic (polar) alignment interaction [58] $\alpha =\frac{1}{2}+\frac{1}{D}$, where *D* is the dimensionality of the experimental system. These values are in good agreement with the experimental data listed in Table 1.

Many experiments adopt millimeter-scale vibrated particles as models to study their cooperative behaviors, and GNF was very common in these experiments. Narayan et al. presented experimental evidence that an agitated monolayer of rod-like particles could form an active nematic phase and show an apparent GNF phenomenon when the area fraction increased from 35% to 66% [48]. Kudrolli et al. showed that polar vibrated rods would aggregate at the sidewall. In contrast, the spherical particles did not show aggregation at the boundary, suggesting that the shape may significantly influence the collective behavior of active particles. But with the increased vibration amplitude, the distribution of particles becomes more homogeneous for both particle shapes. The density fluctuations inside the container were measured as shown in Figure 3a, which is evidence of GNF [50].

**Figure 3 micromachines-13-00295-f003:**
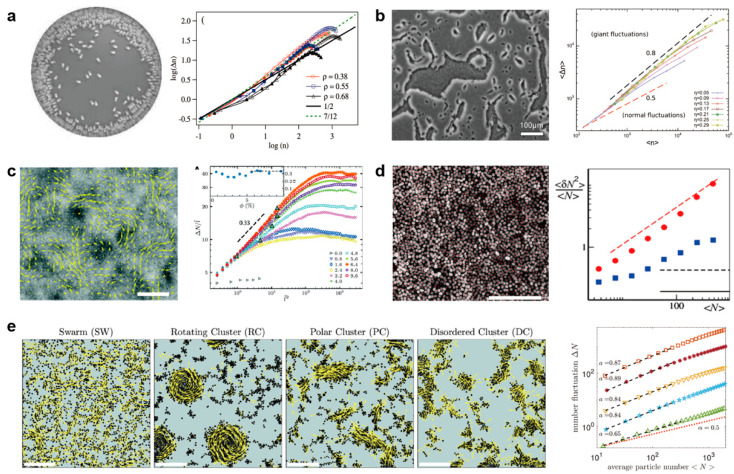
Giant number fluctuation in active matter systems. (**a**) Left: rods migrate and aggregate at the boundaries of a container. Right: the standard deviation of the number of rods Δn versus mean number of rods n inside a circular area at the center of the container. Reproduced with permission from [50]. (**b**) Left: M. xanthus bacteria form moving clustering. Right: the number fluctuation of bacterial at relatively high packing fractions. Reproduced with permission from [52]. (**c**) Left: a snapshot of a dense suspension of E.coli bacteria. Right: fluctuation $\Delta N/\tilde{l}$ as a function of ${\tilde{l}}^{2}$, $\tilde{l}=l/l_{b}$, where l is the side length of selected area, $l_{b}=3\;\mu m$. The scale bar is 85 μm. Reproduced with permission from [53]. (**d**) Left: fluorescence microscopy image of monolayer epithelial cell. The scale bar is 0.5 mm. Right: normalized mean squared number fluctuations. Reproduced with permission from [55]. (**e**) Left: collective states formed by Quincke random walkers with different run and turn times: swarming, rotating cluster, polar cluster, and disordered clusters. The scale bar is 1 mm. Right: number fluctuation in different states. Reproduced with permission from [56].

Bacteria and cells are typical examples of active matter at the micron scale. Zhang et al. investigated the quasi-2D collective motion of wild-type Bacillus subtilis bacteria colonies on agar substrates and reported the first GNF in biological systems [51]. They found that the exponent is quite close to the value of simulation, which is 0.8, and suggested that the statistical properties of collection motion might not depend on the detail of microscopic interaction [60]. Peruani et al. showed collective motion by nonequilibrium cluster formation in *M. xanthus* bacteria when the packing fraction is higher than the critical value, around 17%. The giant number fluctuation characterizes the collective motion without a global orientational order (Figure 3b) [52].

Liu et al. investigated the density fluctuation of bulk *Escherichia coli* suspensions in a 3-dimensional wet (momentum conserving) fluid system and demonstrated the existence of GNF in quasi-3D bacterial suspensions (Figure 3c) [53]. Besides bacteria, cells also show similar properties. The motility of monolayer epithelial cells could be stimulated by the over-expression of a single protein RAB5A, thus promoting the transition to collective migration with enhanced number fluctuation on a large scale (Figure 3d) [55].

In artificial systems, self-propelled colloid particles are the most widely used experimental systems. Karani et al. realized different collective dynamic patterns as showed in Figure 3e by tuning the run and turn times of the so-called Quincke random walkers [56]. In various collective states, the exponent of particle number fluctuation is different but all higher than the situation in the thermal equilibrium value of 0.5 (Figure 3e). Palacci et al. recorded the GNF in the simulation of self-propelled disks, which could form so-called living crystals with similar conditions of their experiments [57].

However, the experiments and theoretical evidence about GNF overwhelmingly prove the existence of GNF. The cause of GNF is still under debate. It may be due to the orientational order and the self-organizing aggregation, but it could also be the consequence of hydrodynamic instability caused by a collective motion of particles [61].

### 2.3. Motility-Induced Phase Separation

Another striking phenomenon in active systems is that self-propelled particles with purely repulsive interactions can experience so-called motility-induced phase separation (MIPS), which is impossible for passive colloidal particles without attraction. To introduce the idea of MIPS, we first consider two limiting models to study the stochastic dynamics of single motile particles (Figure 4a). The first model is run-and-tumble particles (RTP) [62], such as the movement pattern of *Escherichia coli*. This type of particle moves straight with a fixed speed for a period time (relaxation time τ) and suddenly tumble to change the direction with a fixed rate. Another model is the active Brownian particle (ABP) [63], which is similar to how phoretic self-propulsion colloids (e.g., Janus particles) move [64]. The moving direction of particles change smoothly by rotation diffusion, and the trajectory is a continuous curve.

MIPS arises from a simple positive feedback mechanism between particles, such as chemical signaling (e.g., quorum sensing [65]) for microorganisms or steric repulsion for synthetic swimmers, which makes active particles slow down when the local density is higher than average. The reduced motility, in turn, causes the particles to agglomerate [14,66].

Tailleur and Cates first studied a one dimension RTP model and predicted that steady phase separation would occur [67]. Their simulation result proves the existence of spinodal-like phase separation. Later, this finding was extended to the ABP model, and they found that RTP and ABP are equivalent when the time/length scale is large enough and the motility parameter depends on density rather than moving direction [68]. Most importantly, they found that although active and passive phase separation are fundamentally different, coarse-graining of active dynamics at a large scale could map active phase separation to the passive one under certain circumstances for both RTP and ABP. Based on the continuum theory of equilibrium phase separation, Cates et al. proposed active model B and predict the MIPS in 2D [69].

**Figure 4 micromachines-13-00295-f004:**
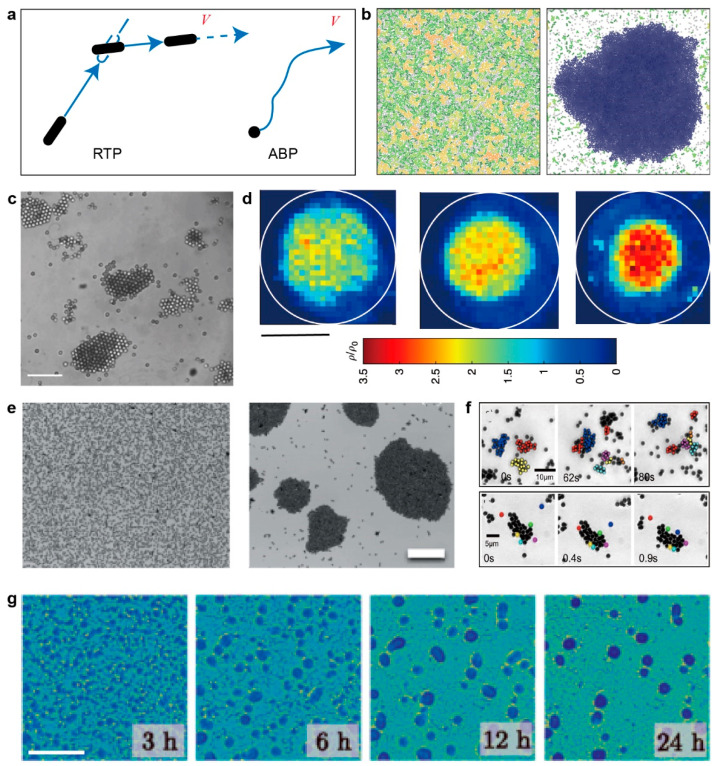
Physical model, simulation, and experiments related to MIPS. (**a**) Schematic of run-and-tumble particle (RTP) and active Brownian particle (ABP). (**b**) Separated phase of solid-like and gas phases without alignment; the packing fraction is 0.7. Reproduced with permission from [70]. (**c**) Phase separation of laser-activated colloidal particles into a few big clusters and a dilute phase. The scale bar is 20 μm. Reproduced with permission from [71]. (**d**) Experimentally measured normalized particle density in steady states for increasing concentration threshold from left to right. The scale bar is 65 μm. Reproduced with permission from [72]. (**e**) Aggregation process in a system of induced-charge electrophoretic self-propelled Janus colloids. The scale bar is 100 μm. Reproduced with permission from [73]). (**f**) Dynamic clusters of self-propelled gold colloidal particles for surface fraction ϕ≈5% and $C_{H_{2}O_{2}}=0.1\%$. Reproduced with permission from [74]. (**g**) Microscopy image of *M. xanthus* cells undergoing phase separation. The scale bar is 0.5 mm. Reproduced with permission from [75].

The simulation of self-propelled soft polar disks on a 2D substrate with isotropic repulsive interaction shows that the cluster formed far away from close packing and without alignment (Figure 4b) [70]. Thus, this simulation result cannot be explained by a hydrodynamic model, which requires orientational order [15]. However, the phase separation theory fits very well with the clustering phenomenon, which occurs in self-propelled particles.

The clustering (also called swarming by some researchers) phenomenon has been observed in many experiments such as self-propelled colloidal particles and bacteria, as listed in Table 2.

Some of these experiments are designed to verify the existence of MIPS, in which the attraction between particles is minimized. For example, Buttinoni et al. studied quasi-2D spherical colloidal carbon-coated Janus particles propelled by the demixing of water-lutidine, caused by light-induced heating on the carbon-coated hemisphere [71]. The attraction in this experimental situation is largely reduced, so the pairwise interaction is mainly repulsive. Under this circumstance, they found the formation of big clusters (Figure 4c). Bechinger’s group used a similar system, but the light-activated particles were controlled individually by a scanning laser system with a feedback loop [72]. They set a threshold of density; if the local density is higher than that value, then the particle will become non-motile particles. In this way, the particles will aggregate into a cluster, and the phase separation occurs (Figure 4d).

Most experiments that show the aggregation of microparticles are not designed specifically to prove MIPS, and in these experiments, the attraction between particles exists. However, MIPS seems to be one of the plausible mechanisms for the aggregation. For example, the induced-charge electrophoretic self-propelled titanium-coated Janus colloids were found to aggregate into clusters when adjusting to the appropriate electric-field frequency [76]. However, this aggregation does not result in complete phase separation, because the clusters start to break apart when their size is beyond a certain threshold. It is argued that this interrupted MIPS may be due to the competitive mechanism of MIPS, structural ordering and polar alignment (Figure 4e) [73]. Theurkauff et al. used platinum-coated gold colloidal particles in hydrogen peroxide as active particles. They found the emergence of dynamic clustering of self-propelled particles when the density is high enough, and the clustering size has a positive correlation with the activity, determined by the concentration of hydrogen peroxide (Figure 4f) [74]. Similarly, Palacci et al. used a suspension of synthetic photoactivated colloidal particles, and they observed that when the blue light is on, homogeneously distributed particles began to assemble into clusters with an average size of 35 particles (~10 μm). They claim that the osmotically driven motion and steric hindrances (collisions) are necessary for the formation of “living crystals” [57].

However, it is not just the self-propelled colloidal particles; the clustering phenomenon is also common in living microorganisms. For example, the bacteria *Myxococcus. xanthus* shows that the formation of fruiting bodies may be partly due to the MIPS because the bacteria could tune their motility based on local density over time [75]. Moreover, since the bacterial motility decreases with density, the phase separation of two bulk phases with different density could form (Figure 4g) [82].

Overall, although these clustering phenomena may be partially attributed to MIPS, the complex interactions (attractive, repulsive, hydrodynamic, quorum sensing, etc.) make the connection elusive. Separating the effects of different interactions remains a challenge.

### 2.4. Relationship between Information and Order

MIPS, as an intrinsic property of many active matter systems, is the result of self-organization. Analogous to the self-assembly in equilibrium systems, self-organization represents a dynamic process of pattern transitions from disorder to order. Such transitions occur across all length and time scales, from molecular to colloidal to macroscopic biological systems. While transitions in equilibrium systems can be characterized by order parameters [83], transitions in non-equilibrium systems lack such a measure. By borrowing insights from information theory, Martiniani et al. have proposed the use of compression-based information entropy to characterize the transitions in dynamic patterns generated in simulations [84].

We have recently started to explore the abstract notion of information and its concrete manifestation in driven self-organizing systems via a combination of theory, simulation, and experiment (Figure 5) [85]. The experimental system we use is built upon our previous work on the dynamic and programmable self-assembly of spinning micro-disks at the air–water interface [86,87,88]. This system could generate a diverse range of patterns (Figure 5a,e), thereby providing an ideal model to study how any given measure-based information varies with patterns. We found entropy by neighbor distances *H_NDist_*, calculated from the distribution of distances between nearest neighbors (defined by Voronoi tessellation) (Figure 5b), to be a very sensitive measure of pattern changes (Figure 5e). On the one hand, the theoretical modeling based on 1D Hamiltonian derived from mean field approximation of pairwise interactions provided an indirect link between information and patterns via pairwise interactions (Figure 5b). On the other hand, the 2D Monte Carlo simulations reproduced the order of the patterns using the distributions of neighbor distances, which provided a direct link between information and order (Figure 5c,d). We expect that the idea of neighbor distance as an information-bearing variable for self-organizing patterns may open up new directions to understand how information entropy relates to other properties of active matter system.

**Figure 5 micromachines-13-00295-f005:**
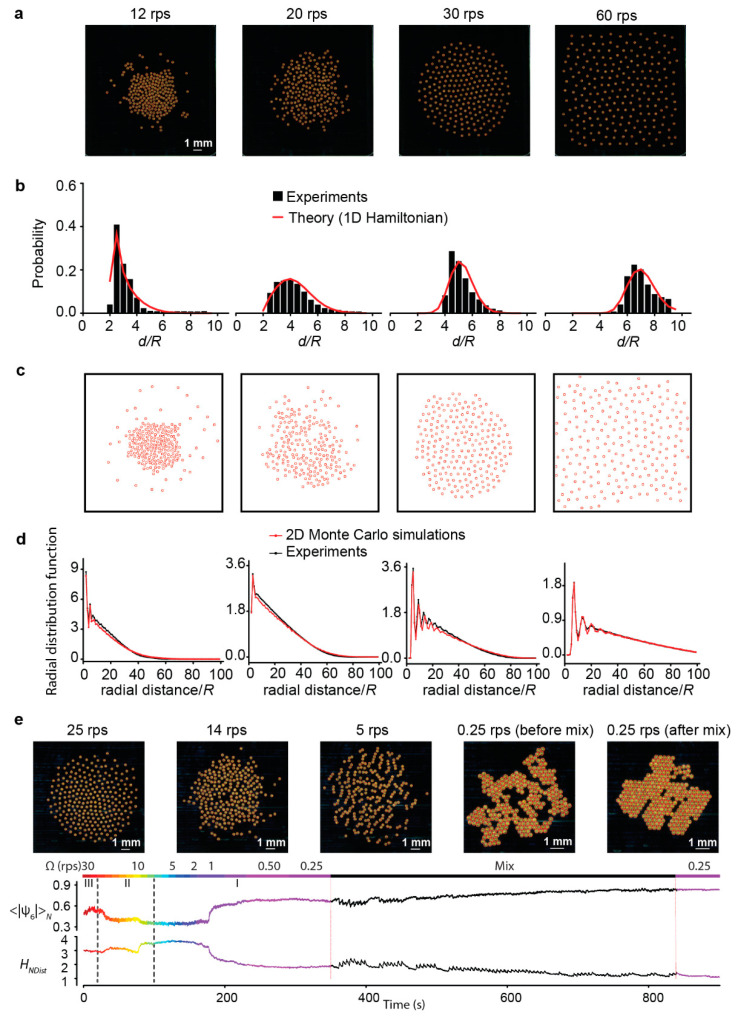
Order and information of a driven self-organizing system consisting of spinning micro-disks at the air-water interface. (**a**) Representative patterns at four different spin speeds. (**b**) The corresponding probability distribution of neighbor distances. The theoretical curves were calculated from an effective Hamiltonian based on mean-field theory. (**c**) 2D Monte Carlo simulation of the patterns based on the information of the neighbor distance distributions in (**b**). (**d**) The radial distribution functions of both experimental patterns and the patterns obtained from the Monte Carlo simulations. (**e**) Tiling experiment that demonstrates the close relation between order, represented by hexagonal order parameter $\psi_{6}$, and information, represented by the Shannon entropy by neighbor distance *H_NDist_* (adapted from [85]).

## 3. The Perspective of Application

The applications of microscopic swarms have been extended to many areas and have undergone significant development in recent years. Different control methods are used for different fields of application. In this section, we will briefly summarize the control and manipulation methods of microscopic swarms and then describe the applications of microscopic swarms in biomedicine and clean environments.

### 3.1. Control and Manipulation

The methods to control and manipulate microscopic swarms can be broadly classified into magnetic, electrical, acoustic, optical, and chemical methods (Figure 6). Each of them has its own advantage and disadvantage in a given application. In this section, we will discuss the pros and cons of each method in the context of biomedical and environmental applications, and our focus is on swarms rather than on individual microrobots.

A swarm contains many individual units, with each unit having several degrees of freedom, and when controlled as a whole, it is an underactuated system. In such systems, the loss of a few units, due to malfunctioning, defects, or other factors, will generally have no significant influence on the functions of the whole swarm, which makes the swarm system robust [89].

All methods of control and manipulation can, in principle, produce forces that aggregate individual microrobots to form swarms, but their effectiveness varies. Magnetic and electric forces can easily change the orientation of a single microrobot and make swarms form regular shapes such as chains and ribbons, thereby making these swarms reconfigurable [76,90]. Acoustic controlling can also produce reconfigurable swarms, but with less variety [91]. Optically- and chemically-driven microrobotic swarms mostly have only limited practicability because of the lack of effective control strategy [92]. One way to overcome this limit is to combine optically- or chemically-driven swarms with other control methods such as acoustic field.

Magnetic control is the most widely used method in controlling microrobotic swarms. This method uses the magnetic force applied on the magnetic microrobots in alternating [93] or gradient magnetic fields generated by coil systems, or by permanent magnets [94]. Because the magnetic interactions are applied at a distance and allow the microrobots to work in confined and narrow spaces, they are suitable to work inside the human body. For the safety aspect, the pathogenicity of the magnetic field is vital in biomedical applications. So far, no clear evidence indicates magnetic fields will endanger patient health when exposure is short. Only when the magnetic field is very strong, i.e., ~10T, some temporary discomfort will occur, such as metallic taste and dizziness [95]. As for the environmental applications, the energy consumption to generate the magnetic fields on a large scale and for a long time using electromagnetic coils is significant and could potentially hinder development in this area. Helmholtz coils are for generating magnetic fields in small spaces. Merritt and Ruben coil systems are capable of generating magnetic fields in a large volume but they are limited to one dimensional fields, and the construction of these coils is also complex and difficult [96]. Besides, simply air-cooling systems is not enough for large systems. An additional cooling device is required for the stable functioning of large coil systems, which require more energy and a complex design.

Electrically-driven microrobots are controlled by the Coulomb force generated by the external electric field. The field will polarize the material and induce dipole within the microrobots. The interactions between dipoles contribute to the swarming behavior of the system [76]. Compared with the magnetic field, the electric field shares similar advantages in working environments. However, for biosafety concerns, living cells can be damaged by the electrophoresis and hydrogel polymerization caused by electric fields [97]. The generation of the electric field has more problems to overcome. Generating electric fields in a large volume requires large electrodes, which are made of more conductive material than magnetic coils. The large electrodes will bring design challenges about safety and efficiency.

Acoustically-driven microrobots are controlled by the energy conveyed by the sound wave, and the microrobots vibrate following the sound wave. By adjusting the phase and frequency of the acoustic wave, researchers can design the acoustic pressure distribution in the system, and thus the microrobots will move according to the pressure gradient and its geometry shape. Unlike magnetic fields, the transmission of sound waves needs a medium [98]. It can bring benefits by designing unique transmission media for a better control effect. It can also lead to problems dealing with complex media, such as the human body. Acoustic control shares the same advantages about the working space and health risk, but the device of acoustic control is much lighter. The feature of microrobots controlled by acoustic waves does not have a strict limitation. This high adaptivity also helps the acoustic field cooperate with other fields.

Optic-controlled microrobots use light to control the movement of microrobots. These microrobots are made from light-responsive polymers, biological materials, or a combination of both. It is easier to make optic-controlled microrobots with better biocompatibility because these materials are softer and closer to biological materials [99]. The light control method has very high accuracy, and the energy transfer efficiency is high when the microrobots are directly exposed to the light source. Properties that can be altered to control the microrobots include the frequency, polarization, pattern, and intensity of light. However, this method could face difficulties when working in internal and complex environments. The penetration ability of light is weaker than the magnetic field. The fiber optic can be used to help light overpass the obstacles [99]. The application of optic control in controlling microrobot swarms is still not as mature as magnetic control and still needs further research in this field. Further research is required for applying optics into swarm control.

Chemically-driven microrobots use the energy produced in chemical reactions to power the locomotion of the microrobots. The microrobots usually function as a catalyst. For example, Au/Pt nanorods are used as the microrobots and catalyze the decomposition of hydrogen peroxide [100]. For chemical actuation, the microrobot will be active as long as the fuel reagents exist. This method is good at powering the microrobot, although it is difficult to control the swarm behavior in real time because the properties of the solution cannot be instantly adjusted. Thus, additional fields such as magnetic and acoustic fields need to be applied to coordinate the swarming behavior of the microrobots. In biomedical applications, the selection of the chemical reaction is vital to the system’s performance. The reaction needs to be safe inside the body, so it is best to use the biochemical reactions that exist inside the human body. Making microrobots use enzymes or ATPs as energy source are good choices [101,102]. In environmental application, however, potential pollution by microrobots themselves is a matter of concern in selecting the proper reaction to power the microrobot.

Currently, the working environment of the microrobot swarms includes solid surface [103], liquid-air [104], liquid-liquid interfaces [105], and inside bulk liquid environment [106]. These environments cover most of the applicational fields of microrobotic swarms. However, the microrobotic swarms working in gaseous environments do not draw much attention. In the biomedical field, we suppose it to be used for curing respiratory diseases. We also envision that the air pollution problem is another possible area of application for such microrobotic swarms. It is conceivable to use aerosols as microrobots and to develop a swarm system for use in a weightless environment such as a space station.

### 3.2. Biomedical Application

Microrobots have been used in the biomedical area because they are small, can be controlled wirelessly, and can reach areas where other methods relying on bulky devices are impossible or hard to reach (Figure 7). However, because of the small mass and volume of a single microrobot, its functions are limited. As a result, researchers have started exploring the use of microrobotic swarms to achieve desired biomedical applications. Grouping many microrobots into a swarm increases the efficacy of intended biomedical application, and controlling a swarm as a whole helps avoid off-target accumulation [23]. In the following section, we will describe three main areas where microrobotic swarms have recently made a significant impact: targeted therapy, in vivo imaging, and biofilm removal.

Targeted therapy has been the main focus of the application of microrobotic swarms in biomedicine. In general, the drugs used to treat diseases, especially cancer, are lethal to both the target tissues and regular tissues and have strong side effects. Targeted therapy assisted with microrobots can deliver agents to the target area and maximize the therapeutic effect while reducing side effects. Nelson et al. used a swarm of helical microrobots, surface-functionalized with near-infrared probes that can be tracked in vivo, to realize magnetically controlled navigation in a mouse [111]. This work initiated the research of microrobotic swarms for targeted therapy. Xie et al. presented the pattern generation and motion control of a snakelike magnetic microrobotic swarm for drug delivery in curved and branched narrow channels, and proved its potential for efficient drug delivery in vivo. The application of microrobotic swarms may reduce side effects. Sitti et al. proposed multifunctional bacteria-driven microswimmers consisting of *E. coli* and magnetic nanoparticles for targeted drug delivery to cancer cells [107]. Fischer et al. reported the micropropellers that show controlled locomotion as a swarm under the magnetic field through the eyeball of porcine [116]. Its surface is treated with fluorocarbon to reduce the resistance in the vitreous fluid of the eye, enabling the micropropellers to penetrate the vitreous humor and reach the retina for the first time.

In targeted therapy, the cargos being transported can be conventional biological or chemical drugs, as in the examples in the previous paragraph or other forms such as the reagents used for brachytherapy, hyperthermia, and thrombolytic therapy. Brachytherapy needs to place a radioactive source close to the targeted tissue to destroy the tumor cells [117]. Compared with traditional radiotherapy, it is less harmful to the human body. Microrobots can help with the placement of the radioactive reagents through controlled navigation. At present, there are relatively few applications of single microrobots in brachytherapy [109], and further research associated with microrobotic swarms is needed. Hyperthermia uses the characteristic that tumor cells are more sensitive to high temperatures (typically in the range of 40–45 °C) than normal cells to selectively destroy them [118]. The often-used methods for wireless heat delivery include high-frequency magnetic fields and ultrasonic-resonance of mechanical structures [119]. The local heat source can be microrobotic swarms. Wang et al. utilized reconfigurable swarms of ferromagnetic colloidal particles for enhanced local hyperthermia in medical oncology [110]. Although this experiment only shows the inhibition effect on tumor cells in vitro with high precision (up to the micrometer scale), it offers a new method for the localized treatment of tumors. In thrombolytic therapy, microrobotic swarms can apply mechanical force and increase the flow rate to help the blood clot dissolve. A tissue plasminogen activator (t-PA) is a kind of thrombolytic drug, which is easy to spread to the whole body and induces side effects such as symptomatic intracranial hemorrhages (SIH), limiting its use. Manamanchaiyaporn et al. reported a kind of magnetite nanoparticle swarm, which can capture the t-PA in the magnetic field and transport it to the location of the blood clot [120]. Guo et al. used the phase transition of perfluorohexane, a nonpharmaceutical treatment strategy, in their nanoparticles to directionally destroy the thrombus tissue, which can avoid the side effects of the traditional pharmaceutical treatment strategy to thrombus [108].

Apart from targeted delivery, the swarms of microrobots can increase the signal-to-noise ratio for in vivo imaging and in turn be guided by in vivo imaging. In vivo imaging of microrobotic swarms is crucial because it not only helps to achieve remote control of microrobotic swarms, but also helps locate the target tissue in vivo. The imaging has been realized by different methods in recent years, including fluorescent imaging (FI), ultrasound imaging (UI), magnetic particle imaging (MPI), and magnetic resonance imaging (MRI). Nelson et al. used NIR-797 as fluorophore to realize fluorescent imaging of the swarm in the intra peritoneal cavity of mice [111]. Zhang et al. reported the navigation of a magnetic nanoparticles-based robotic swarm by ultrasound imaging [103]. Zhang et al. also reported in vivo magnetic resonance imaging of a swarm of microswimmers inside the stomachs of mice to track the location [121].

In addition to in vivo applications, microrobotic swarms can also be used to remove harmful biofilms. For example, they have been used to destroy membranes of bacteria and remove biofilm accumulated on medical and industrial equipment. Koo et al. created catalytic antimicrobial robots assembled by catalytic–magnetic nanoparticles to remove biofilms [122]. Zhang et al. designed a porous magnetic microswarm to eliminate biofilms through synergistic effects of the chemical and physical process [112]. As a result, they provide another convenient method to clean medical and industrial equipment.

Although the application of microrobotic swarms in biomedicine has made significant progress over the past few years, there are still significant challenges. For example, the efficiency of targeted therapy using microrobotic swarms can be low. In one report, less than one percent of the drugs were delivered to solid tumors in cancer treatment [123]. Part of the reason for the inefficiency is the immune response of the human body. The microrobots are recognized as intruders and can be eliminated by immune cells. In addition to the issues of low efficiency of delivery and biocompatibility, other limitations include the lack of intelligent control methods, the relatively low quality of in vivo imaging, and the lack of system integration of swarms in clinical settings [23]. These issues need to be solved in future research.

### 3.3. Environmental Application

Microrobotic swarms have also been used in the environment area, particularly in the area of clean water. Heavy metal pollution, oil leakage, and industrial sewage all contribute to the water shortage problem. Existing methods may not meet demands. Although single microrobots have been applied in environmental treatment for a few years because of their superior micro/nanoscale effects and the active motion that accelerates the diffusion-limited processes [23,124,125,126,127], the application of microrobotic swarms in this area has just begun.

Xie et al. developed magnetic microsubmarines based on sunflower pollen grains [114]. The porous structure can efficiently remove the leaked oil and plastic microparticles in polluted water. Zhang et al. fabricated porous spore biohybrid adsorbents combined with magnetic-driven microrobots to enhance the adsorption capacity and the magnetically-propelled locomotion, which used a biological hybridization method to treat heavy metal-containing sewage [115]. Pumera et al. designed light-driven microrobots of photocatalytic materials, which show swarming behavior under light irradiation to degrade disposable textiles [128]. Pumera et al. created an adhesive polydopamine magnetic microrobot inspired by the natural mussel to remove microplastics in water, and the swarm of these microrobots possesses more significant potential to achieve removal [129]. Simmchen et al. presented photocatalytic Au@Ni@TiO2 micromotors to eliminate microplastic in water. The single particles can also assemble into long chains to enhance the effect of the treatment [113].

Although microrobots play an increasingly important role in solving environmental issues, the use of microrobotic swarms is still relatively new and needs further development. If misused and mismanaged, they may also become a source of pollution [130]. Some self-propelled microrobots contain multiple heavy metals. When they complete their tasks, they may disperse heavy metal into the surrounding environment due to electrochemical corrosion, thereby becoming a source of pollution themselves [131]. The locomotion of these toxic microrobots may lead to the further spread of pollution. Therefore, the development of bio-friendly and environmentally friendly microrobots should be considered for future research [129].

## 4. Summary and Future Perspective

From the perspective of fundamental physics, active matter systems are partially understood under specific conditions. Active agents such as self-phoretic colloidal particles and swimming bacteria exhibit non-equilibrium behaviors such as long-range order, GNF, and MIPS. These behaviors distinguish them from their equilibrium counterparts. The observation of these behaviors in experiments has provided evidence to support the existing physical models. In the long term, researchers hope to build a generic model to describe all types of active matter systems, including living and non-living systems, just as the Ising model in equilibrium systems [132]. Equipped with this further information, we could harness the unique properties and functions of active matter in more areas of applications.

From the perspective of application, microscopic swarms have experienced rapid development and proved to have potential applications in biomedicine and clean environments. Nevertheless, most of the current results are proof-of-concept demonstrations, and the applications in clinical settings or in real environments require further research efforts. Moreover, the properties and functions of most of the current microscopic swarms are simple aggregates of the effects of individual units. It is still a challenge to realize, given the emergent function of a swarm through the adjustment of local interactions and dynamic cooperation between individuals in ways that mimic collective phenomena in biology.

More importantly, through this review, we have identified an opportunity to bridge the gap between the community of active matter physics and the community of microrobotics. The purpose of this review is to combine these two perspectives and encourage more researchers to think about the relationship between the fundamental physics of active matter and the application of microrobotics. On the one hand, the unusual behaviors and properties discovered in the active matter system may have great application potential in controlling the behaviors of microrobotic swarms. Utilizing these behaviors and properties may guide us in controlling the collective motion of microrobots, thus achieving more functions. For example, understanding the clustering mechanisms in MIPS could help control the vastly underactuated swarms as a whole. On the other hand, microrobotic swarms can be regarded as a form of active matter system and be used to verify predictions by theories and numerical simulations, and possibly even trigger us to investigate other new laws related to the nonequilibrium statistical physics. Future research in the interdisciplinary area between active matter physics and microrobotics (micromachines) will develop our understanding of collective motion phenomenon in nature and our ability to apply microscopic swarms in the medical, environmental, and many other fields.

## Figures and Tables

**Figure 1 micromachines-13-00295-f001:**
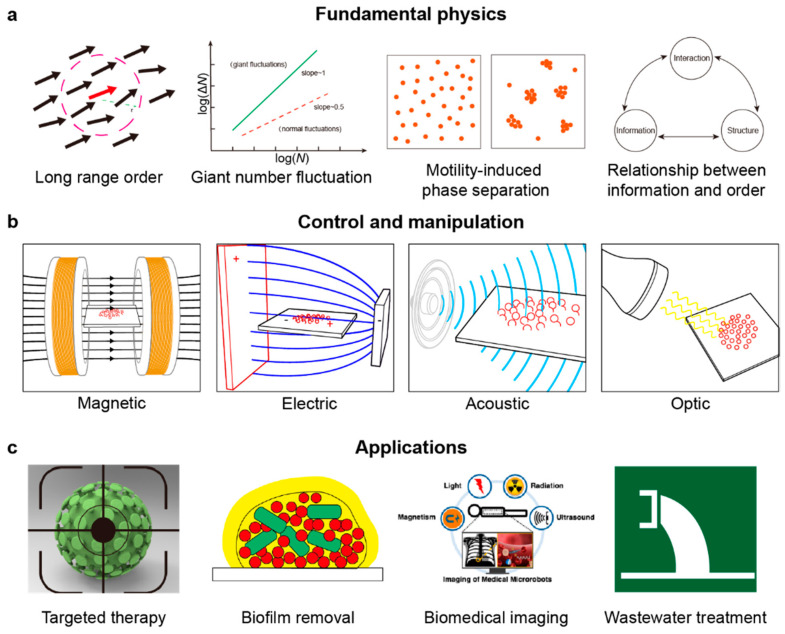
Overview of the perspective of fundamental physics, manipulation, and applications of microscopic swarms. (**a**) Schematic representing fundamental physics of active matter systems. (**b**) Schematic of different control and manipulation methods. (**c**) Application fields of microscopic swarms in biomedicine and the environment.

**Figure 2 micromachines-13-00295-f002:**
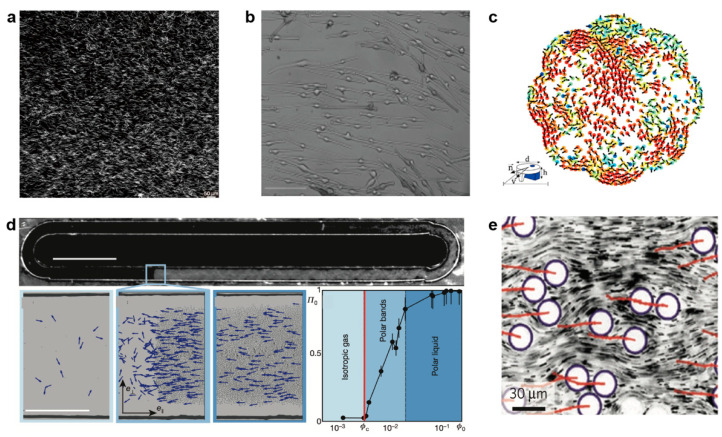
The existence of long-range order in microscopic active systems. (**a**) Nematically ordered phase in a thin fluid layer of filamentous bacteria at high density. Reproduced with permission from [36]. (**b**) Long-range oriented human melanocytes cells. Scale bar, 100 μm. Reproduced with permission from [38]. (**c**) Collective motion of self-propelled disks; bottom left panel: a sketch of the polar disk with diameter d = 4 mm. Reproduced with permission from [39]. (**d**) Top: dark-field images of roller particles that spontaneously form a microscopic band propagating along the channel. Scale bar, 5 mm. Bottom images show enlarged views of different phase: isotropic gas; polar bands; polar liquid. Scale bar, 500 μm. Reproduced with permission from [40]. (**e**) The microfluidic 2D droplet ensemble and velocity fluctuations. Reproduced with permission from [41].

**Figure 6 micromachines-13-00295-f006:**
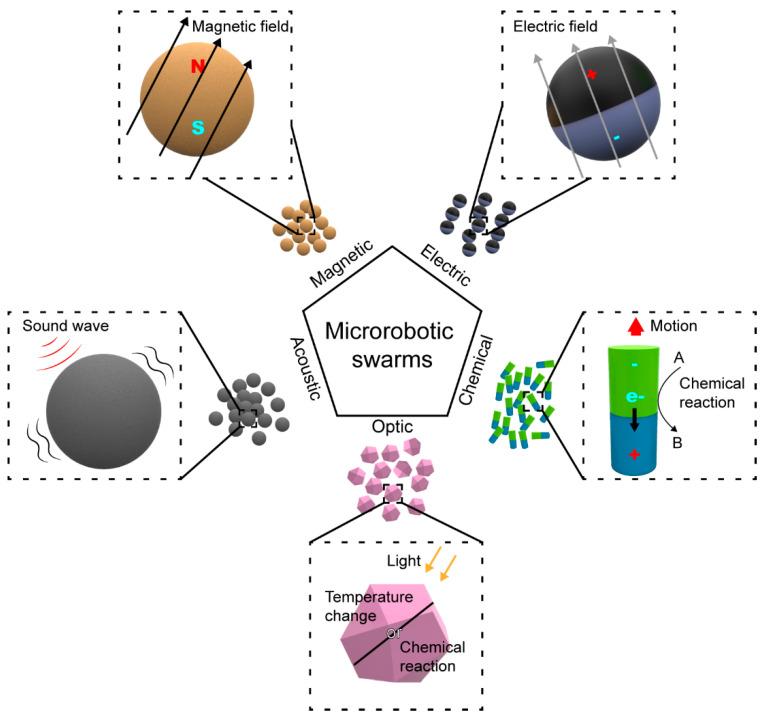
Actuation mechanisms of different methods of control and manipulation.

**Figure 7 micromachines-13-00295-f007:**
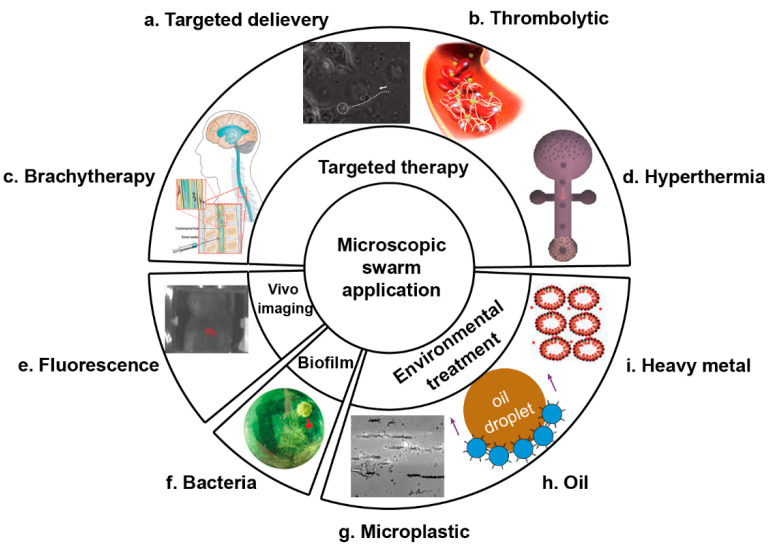
Schematic illustration of microscopic swarm application. (**a**) Targeted delivery of bacteria-driven microswimmers under directional guidance toward cancer cells. Reproduced with permission from [107]. Copyright 2017 American Chemical Society. (**b**) Nanoparticles used in phase transition thrombolysis. Reproduced with permission from [108]. Copyright 2019 American Chemical Society. (**c**) Helical microswimmer in brain brachytherapy. Reproduced with permission under a Creative Commons CC-BY license from [109]. Copyright 2020 MDPI. (**d**) Locomotion of a ferromagnetic colloidal swarm toward the tumor cells in hyperthermia. Reproduced with permission from [110]. Copyright 2018 Wiley-VCH GmbH & Co. KGaA, Weinheim. (**e**) Fluorescence in vivo imaging of f-ABFs in mice. Reproduced with permission from [111]. Copyright 2015 WILEY-VCH Verlag GmbH & Co. KGaA, Weinheim. (**f**) Removal of bacteria biofilms by the p-Fe3O4 MPs swarm. Reproduced with permission from [112]. Copyright 2021 American Chemical Society. (**g**) Magnetic Au@Ni@TiO2 chains remove microplastic sample under the magnetic field and UV light in 0.2% H2O2 solution. Reproduced with permission from [113]. Copyright 2019 American Chemical Society. (**h**) Cooperative microsubmarines for oil removal in water remediation. Reproduced with permission from [114]. Copyright 2020 Elsevier Ltd. All rights reserved. (**i**) Porous spore@Fe3O4 biohybrid adsorbents adsorb and remove heavy metal ions. Reproduced with permission from [115]. Copyright 2018 Wiley-VCH GmbH & Co. KGaA, Weinheim.

**Table 1 micromachines-13-00295-t001:** Giant number fluctuations (GNFs) phenomena in experiments; d = diameter, l = length.

Type of System	Model (Characteristic Size)	Exponent α	Reference
Millimeter-scale particles	Granular rods (d = 0.8/l = 4.6 mm)	1	[48]
	Spherical particles (d = 1.0 mm)	1	[49]
	Polar granular rods (d = 4.8/l = 9.5 mm)	0.66	[50]
	Polar disks (d = 4 mm)	0.8	[39,42]
Bacteria	Cylindrical Bacillus subtilis (l = 5.0/d = 1.0 µm)	0.75	[51]
	Rod-shaped Myxococcus xanthus (l = 6.3/d = 0.7 µm)	0.85	[52]
	Filamentous Escherichia coli (l = 20/d = 0.8 µm)	0.63	[36]
	Escherichia coli in quasi-3D System (l=3.0/d=0.65 µm )	0.83	[53]
Cells	Neural progenitor cell (l = 100/d = 10 µm)	0.75	[54]
	Flocking epithelium (d = 30 µm )	0.8	[55]
Colloidal particles	Quincke roller colloid (d=40 µm)	0.85	[56]
	Photoactivated colloid (d = 1.5 µm)	0.9	[57]

**Table 2 micromachines-13-00295-t002:** Experiments showing clustering phenomena; d = diameter, l = length.

Type of System	Model (Characteristic Size)	Primary Swarming Mechanism	References
Artificial system (Near-zero attractive interaction)	Light-activated carbon-coated Janus particles (d = 4 µm) in near-critical water-lutidine mixture	MIPS	[71,72]
Artificial system (significant attractive interaction)	Light-activated polymer (TPM) sphere (d = 1.5 µm)	Osmotically-driven motion and collision	[57]
	Silica Janus particles half coated with titanium (d=4 μm) in deionized water controlled by a.c. electric fields	Induced-charge electrophoresis	[73,76]
	Ir/SiO_2_ Janus particles (d = 1.2 µm) with low level of hydrazine	Diffusioosmotic Diffusiophoresis	[77]
	Au particles (d = 1 µm) in H_2_O_2_ solution spiked with hydrazine	Diffusiophoresis Diffusioosmosis Electrophoresis Electroosmosis	[11]
	Spherical gold colloids half covered with platinum in H_2_O_2_ solution (d = 1 µm)		[74]
	UV-activated AgCl particle (d = 1 µm) in deionized water		[78]
	Ag_3_PO_4_ microparticle (d = 2 µm) schooling controlled by addition or removal of NH_3_		[79]
Biological world	Bacteria *Myxococcus xanthus* (l = 5 µm)	Quorum sensing/ Chemotaxis	[75]
	Bacteria *Dictyostelium discoideum* (l = 20 µm)		[80,81]

## Data Availability

Not applicable.

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
