# Peer review of "Microscopic Swarms: From Active Matter Physics to Biomedical and Environmental Applications"

_micromachines, 2022, doi:10.3390/mi13020295_

Round 1

Reviewer 1 Report

Comments to the Authors:

In summary:

Authors summarize a comprehensive and profound review about the microscopic swarms, from fundamental active matter physics like long-range order, giant number fluctuation, motility-induced phase separation and information-order relationship, to current control and manipulation like magnetics, electrics, acoustics and optics, to biomedical and environmental applications. This review is a good bridge to link the community of active matter physis and the community of microrobotics, and to inspire fruitful collaborations between the two communities.

Before I recommend this manuscript for publication in Micromachines, several issues should be clarified, listed as below:

  1. For the Giant Number Fluctuation,
    1. The y-axis of the graph in the Figure 1a should be corrected as [log (√N)base of logarithm (ΔN/√N) ]rather than current ΔN/√N, and the green line should be curved rather than straight, if the value is 1 and 2 in y-axis. Or the authors can change the current linear scale of y-axis to a log scale.
    2. To be accurate for the equilibrium system, the fluctuation of ΔN/√N should be weaker and weaker as the N increases.
  2. For the Long-range order, it may not be good enough to just collect the previous research results. It would be more profound to address the cause of these long-range-order phenomena in Section 2.1, just as the brief description of the cause of GNF in the last paragraph of Section 2.2.

  1. Minor corrections:
  2. In lines 124, “Vchanges” should be “(V) changes”.
  3. In lines 141 and 256, “d=dimeter” should be “d=diameter”. The “d” here is mixed with “d” in line 138 and 139, for “dimensionality” is different from “diameter”.
  4. In line 186, the ~ on the l.
  5. In line 237, “In this the way” should be “In this way”.
  6. In line 484, “the in vivo tumor cell inhibition” is not a grammatical expression.
  7. In line 659, something wrong with the transfer from latex to the pdf about “$\mathrm{XY}$” in the reference 27.

Reviewer 2 Report

In the manuscript, the authors present a review about microscopic swarms: from active matter physics to biomedical and environmental applications.

The topic discussed is presented in a compact and interesting form. The literature review and the issues surrounding this topic have been studied in depth. To my knowledge, no article deals with this interesting topic in such a way. The authors present the possibilities of filling the gap between the community of active matter physics and the community of microrobotics.

The manuscript has sufficient scientific quality and relevance for Micromachines in this form. I suggest accepting in the present form (I recommend checking it by nativespiker).
